# Battery State-of-Health Estimation: A Step towards Battery Digital Twins

Vahid Safavi [1,*,†], Najmeh Bazmohammadi [1,†], Juan C. Vasquez [1,†] and Josep M. Guerrero [1,2,3,†]

1 Center for Research on Microgrids (CROM), AAU Energy, Aalborg University, 9220 Aalborg East, Denmark; naj@energy.aau.dk (N.B.); juq@energy.aau.dk (J.C.V.); josep.m.guerrero@upc.edu (J.M.G.)
2 Center for Research on Microgrids (CROM), Department of Electronic Engineering, Technical University of Catalonia, 08034 Barcelona, Spain
3 Catalan Institution for Research and Advanced Studies (ICREA), Pg. Lluís Companys 23, 08010 Barcelona, Spain
* Correspondence: vsa@et.aau.dk
† These authors contributed equally to this work.

**Abstract:** For a lithium-ion (Li-ion) battery to operate safely and reliably, an accurate state of health (SOH) estimation is crucial. Data-driven models with manual feature extraction are commonly used for battery SOH estimation, requiring extensive expert knowledge to extract features. In this regard, a novel data pre-processing model is proposed in this paper to extract health-related features automatically from battery-discharging data for SOH estimation. In the proposed method, one-dimensional (1D) voltage data are converted to two-dimensional (2D) data, and a new data set is created using a 2D sliding window. Then, features are automatically extracted in the machine learning (ML) training process. Finally, the estimation of the SOH is achieved by forecasting the battery voltage in the subsequent cycle. The performance of the proposed technique is evaluated on the NASA public data set for a Li-ion battery degradation analysis in four different scenarios. The simulation results show a considerable reduction in the RMSE of battery SOH estimation. The proposed method eliminates the need for the manual extraction and evaluation of features, which is an important step toward automating the SOH estimation process and developing battery digital twins.

**Keywords:** lithium-ion batteries; state of health; data pre-processing; discharging characteristics; digital twin; deep learning; CNN-LSTM



## 1. Introduction

Lithium-ion (Li-ion) batteries are attractive power storage technologies for many applications, such as in power systems, electric cars, telecommunications, aerospace, and other industries, due to their small size, high energy density, low self-discharge rate, low cost, and long life [1]. Li-ion batteries degrade over time and charging/discharging cycles, which results in reductions in their capacity and output power [2]. Also, the incorrect utilization of the battery increases its speed of degradation. A battery management system (BMS) can provide risk warnings to users and help with timely battery maintenance and replacements to guarantee the satisfactory performance of a battery [3]. An essential metric for monitoring the health status of batteries in BMS is their state-of-health (SOH), which is defined as the ratio of the current capacity of the battery to its initial capacity. The SOH directly indicates the degree of degradation of the battery over its lifetime [4]. Accidents such as battery leakage, insulation failure, and partial short circuit problems can occur when battery health degrades to a certain degree and can create several safety issues [5,6]. A battery's SOH cannot be measured directly by sensors; instead, it can only be calculated from measurable variables such as battery voltage, current, and temperature [7]. Additionally, the degradation of Li-ion batteries is a very complex process with varying degrees of dependence on the working conditions [8].

The internal characteristics of Li-ion batteries are highly nonlinear, and their lifespan is influenced by several factors. In this sense, accurate state of charge estimation, SOH monitoring, thermal management, and increasing the useful life of batteries are among the main challenges of BMS [9]. During recent years, digital twin (DT) technology has attracted a great deal of attention for battery monitoring and control from both academia and industry. Using DT, living models can be developed for systems and components to closely track their SOH and completely understand their degradation behavior to prolong components' life and minimize abnormal events [10]. A good overview of DT-based predictive maintenance can be found in [11–13].

Although the DT technique is still in its early stages of development, it has already been shown to be useful in the design, monitoring, and control of complex systems, such as Li-ion batteries. A battery DT creates a mapping between the physical entity and its virtual model, which interact closely with one another through bidirectional data exchange. Big data analytics, artificial intelligence, blockchain, and the (Industrial) Internet of Things are among the main DT-enabling technologies [14]. Figure 1 represents a general overview of a battery DT and its services. Zhao et al. [15] present a DT framework using machine learning (ML) for battery SOH estimation to improve the BMS and optimize the operation of battery storage units. The framework combines a hybrid model that integrates long short-term memory (LSTM) as a data-driven model to provide precise initial state-of-charge (SOC) estimations and impedance data for an extended Kalman filter as a physical-based model. The BMS measures, estimates, and regulates battery conditions to ensure the efficient and safe operation of the battery. However, numerous voltage, current, and temperature sensors are needed, which can pose new challenges, such as high costs, limited space, poor efficiency, and high failure rates. Muaaz et al. [16] suggested a DT-based solution to these issues that allows the BMS to estimate and predict the SOH of the battery using only a voltage sensor. They also explained the importance and feasibility of implementing a BMS DT. In addition, to identify relevant variables for ML modeling, they analyzed a correlation matrix of all the variables of the BMS and considered variables with a correlation coefficient greater than 0.6. Simple and multi-linear regression models were then employed to predict the continuous output variables.

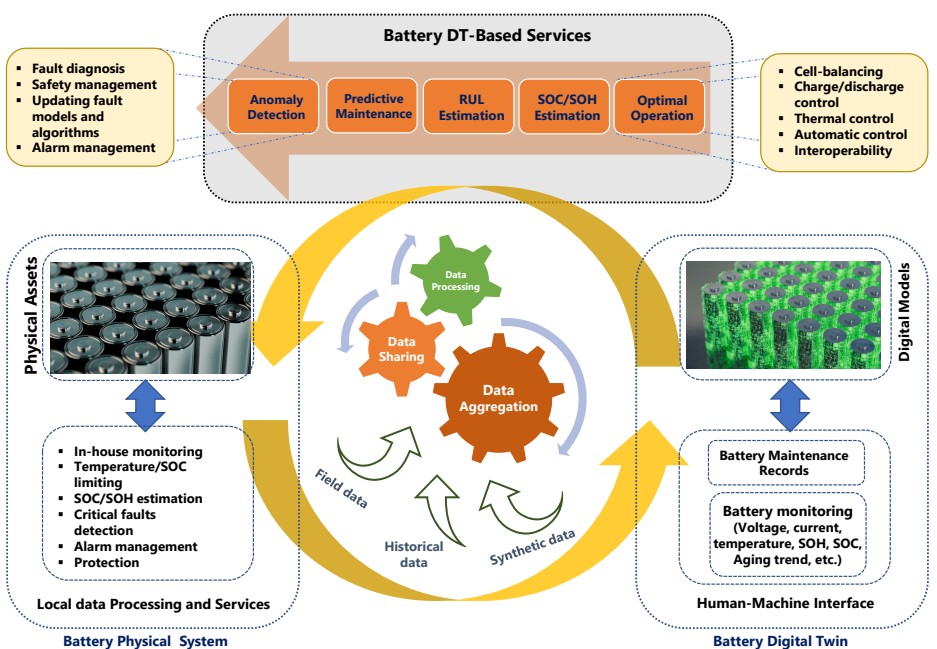

**Figure 1.** A general overview of a battery DT and its services.

One of the most important services of a battery DT is providing an accurate estimation of the SOH of the battery during its lifetime. Such information is critical for the efficient

operational management of the battery, especially when integrating it with other power resources in microgrids, and the timely maintenance of the battery. The role of the battery DT is to provide an accurate model of the battery considering its operating environment as well as an efficient platform for integrating and managing the enormous amounts of data that are collected over the battery's life [17]. The DT-driven SOH estimation strategy features key attributes of autonomy and adaptation. The ultimate goal is to have a SOH estimation technique that is automatically updated based on the most recent information about the battery and the surrounding environment using efficient learning algorithms. Although there are many studies dedicated to battery SOH estimation, a DT-based approach taking full advantage of DT technologies is still missing. To develop such a model, a step-by-step approach should be adopted. This paper provides a brief overview of the current SOH estimation strategies for Li-ion batteries. A SOH estimation method with a novel data pre-processing technique is then proposed and validated through different simulation scenarios to enhance the SOH estimation accuracy without increasing the model's complexity. Afterward, future steps towards developing an adaptive and automatic SOH estimation technique are discussed, and a conceptual framework for battery digital twinning is presented.

The rest of this paper is organized as follows. In Section 2, a brief overview of battery SOH estimation methods is given. Section 3 is dedicated to the introduction of the NASA data set for battery degradation analysis, and feature extraction for predicting the SOH of Li-ion batteries is also discussed. A novel data pre-processing framework for ML-based models to estimate battery SOH is provided in Section 4. In Section 5, the performance of the proposed model is evaluated in four different scenarios, and simulation results are thoroughly discussed. In Section 6, online learning is introduced to develop a DT of a battery. Finally, concluding remarks are presented in Section 7.

## 2. Battery SOH Estimation

Many studies have been conducted to determine an accurate estimation of the SOH of batteries. Generally, these studies can be classified into two categories: model-based and data-driven methods. The three commonly used model-based methods used for SOH estimation are based on equivalent circuit, electrochemical, and empirical models [18]. Modeling batteries with equivalent circuits is a simplistic approach with low accuracy [19]. Electrochemical models are accurate laboratory models with complex partial differential equations [20], while empirical models use a simple model of the battery to identify the key parameters with difficulty in describing the capacity degradation [21]. Nowadays, continuous advancements in enhancing the electrochemical performance of Li-ion batteries are achieved by applying intricate chemical mechanisms that make the physical modeling of Li-ion batteries more and more complicated. Reference [22] introduced a self-assembly strategy focusing on terminal groups to improve lithium adsorption capability, low activation energy for lithium diffusion, and excellent structural stability. On the other hand, data-driven techniques provide appropriate solutions to battery degradation estimation problems with fewer requirements for understanding the physical and chemical mechanisms. Due to their flexibility and effectiveness, data-driven methods have been increasingly attracting interest in the SOH estimation of batteries. In data-driven models, the SOH of the battery is directly calculated using historical data that are easy to collect from sensors. Data-driven models for battery SOH estimation can be further classified into two categories: statistical and ML models [2].

A data-driven battery SOH estimation method generally involves the following four steps: data collection, feature extraction, model training, and SOH estimation [23]. Many data-driven techniques have been presented for battery SOH estimation, which either directly use battery capacity data or extract features from sensor data to estimate the SOH [24]. The remaining useful life (RUL) of a battery defines how many charge–discharge cycles are left before the battery state drops to a specific threshold. To compute the RUL of the battery, Lyu et al. [25] used an optimized relevance vector machine approach. To predict

the RUL and SOH of Li-ion batteries, Khumprom and Yodo [26] provided a data-driven prognostic method using deep Neural networks (NNs); the effectiveness of data-driven methods for battery aging characterization and SOH estimation was found to depend on the quality and quantity of the available data. Sufficient and representative data from various battery operating conditions and aging scenarios are crucial to ensure accurate predictions. By analyzing, identifying, and classifying the patterns and trends of the measurement data, such as the voltage, current, and temperature, and deploying deep neural network (DNN) models, an aging behavior pattern indicator for battery SOH estimation was proposed by Xia and Abu Qahouq [27]. To forecast battery aging, She et al. [28] integrated an incremental capacity analysis with a radial basis function (RBF) NN. The input data, including the accumulated mileage of vehicles and initial charging SOC, average charging temperature, average charging current, and the average discharging temperature of battery systems, and a classification factor for the incremental capacity peak values were chosen. A RBF NN model was used to reduce the dependency on the quality of data sets and increase the flexibility for noisy data while being particularly useful for pattern recognition and regression tasks. It could also learn the relationship between the battery's aging parameters (capacity fade) and the corresponding incremental capacity analysis values and measurement data.

Convolutional neural network (CNN), recurrent neural network (RNN), gated recurrent unit (GRU), and LSTM models for predicting battery SOH and RUL have been established by different researchers for predicting the SOH and RUL of Li-ion batteries. An RNN model was provided in [29] to estimate SOH degradation in batteries using current and voltage measurements. To improve the accuracy of the battery RUL prediction, Park et al. [30] combined multichannel charging profiles with a LSTM model. Zheng et al. [31] proposed a CNN–GRU model using current and voltage measurement data to estimate battery SOH. Ren et al. [32] used a CNN–LSTM model to predict the RUL of the battery by considering the discharge capacity, linearly interpolated discharge capacity, linearly interpolated temperature, the quantity of discharge, and discharge time as main parameters. Kara et al. [33] combined a CNN–LSTM model with particle swarm optimization (PSO) to improve the accuracy of RUL prediction. In this model, spatiotemporal relations are extracted from multivariate time series data, and the PSO algorithm is utilized to optimize the NN hyperparameters.

Data-driven models that do not directly use sensor data, such as voltage, current, and temperature, cannot estimate battery capacity with high accuracy. A solution to this is to extract efficient health features from measured data to improve the performance of SOH prediction models. There are two types of feature extraction: manual and automatic. In the manual feature extraction techniques, features are selected from the charging or discharging data of the battery, such as voltage, current, etc., and the most relevant features are fed into ML algorithms to estimate battery capacity [34]. For example, Cui et al. [35] extracted the duration of the constant voltage (CV) phase, the ratio of the constant current (CC) time and charging time, the area under the CC characteristic, and the time required for equal voltage rise, current drop, and voltage drop intervals as features. Then, the extracted features were evaluated by the Pearson and Spearman's rank correlation coefficients. To evaluate the short-term SOH and long-term RUL of Li-ion batteries, Wang et al. [36] applied indirect health indicators, such as the time corresponding to the same charging voltage interval, discharge voltage interval, and discharge temperature change, and a Gaussian process regression model. Zhu et al. [37] proposed a method to estimate the battery SOH using a support vector machine and extracted features from battery cycle data. They focused on the relaxation process that occurs after a full charge, as it is highly related to battery degradation. Each voltage relaxation characteristic was transformed into six statistical features: the variance, skewness, maxima, minima, mean, and excess kurtosis. These features served as input to the SVM for SOH estimation, allowing for the accurate assessment of battery health based on the extracted characteristics from the relaxation process.

To track battery degradation, Ma et al. [38] integrated a RNN model with the differential thermal voltammetry (DTV) signal processing method. In the beginning, DTV curves were smoothed using the Savitzky–Golay method and Pearson correlation analysis, and three key features were derived from various dimensions. Then, using National Aeronautics and Space Administration (NASA) [39] battery data sets, four RNNs were developed and analyzed. In addition, a Monte Carlo (MC) simulation was performed to evaluate the performance under uncertainty, and the Bayesian optimization method was applied to modify hyper-parameters. A SOH estimation method integrating transfer learning and a deep belief network (DBN)–LSTM model was presented by Ma et al. [40]. Transfer learning was applied to learn the features, and a DBN was used to estimate the SOH of the battery, while LSTM was utilized to consider the impact of historical data. The measured voltage, current, and temperature were used directly to extract six health features. The performance of their proposed technique was evaluated on the MIT battery data set and a mean absolute percentage error (MAPE) of 0.99% was reported.

Xu et al. [41] presented a feature selection method to improve the training efficiency of a NN by eliminating irrelevant features from the input data. Also, by increasing the layers of the NN, the accuracy rate tends to saturate and bring more errors to the prediction results. They addressed this problem by incorporating skip connections into the CNN–LSTM model to improve the accuracy of SOH prediction while reducing the computational burden and enhancing the robustness. A CNN is used to extract features from the original data, LSTM is applied to predict the SOH, and a skip-connection is added by concatenating the outputs of two different LSTMs and using concatenated outputs as an input layer of LSTM.

Manual feature extraction in battery SOH estimation has the merit of being rooted in the understanding of the battery's physical behavior and can be scaled and applied effectively. However, the need for developing efficient automatic feature extraction methods is growing due to the increasing complexity and diversity of battery data sets. Automatic feature extraction offers the advantage of discovering hidden patterns and capturing complex relationships that may not be apparent while manually selecting features. It enables the exploration of a broader range of features and facilitates a more comprehensive characterization of battery behavior. Additionally, automatic feature extraction techniques can adapt to varying battery chemistries, configurations, and operating conditions, ensuring scalability and generalizability [42].

For battery SOH estimation, there are not many studies that specifically investigate feature extraction automatically, and existing techniques frequently combine feature extraction and SOH estimation techniques into a supervised model that is trained using SOH values as labels and raw battery data as inputs. An effective deep learning (DL) model that is frequently used to extract features from measured data automatically is the CNN architecture. Typically, a one-dimensional (1D) CNN is utilized to extract the temporal features of sequence data, while a two-dimensional (2D) CNN is applied to extract spatial features, such as images [43]. Gong et al. [44] calculated battery SOH using a CNN and encoder–decoder model while the encoder combines a CNN with a super-attention method for automatic feature extraction. In their model, for estimating battery SOH, only the raw measurements of battery current, voltage, and temperature are required.

CNN models are commonly used to extract features automatically from sensor data. However, in most studies, the focus is on improving the feature extraction methods to enhance the accuracy of SOH estimation by increasing the model complexity [45] without much attention to the pre-processing of data. Zhou et al. [46] proposed an attention depthwise temporal convolutional neural network (TCNN) model for feature extraction to improve battery SOH prediction. The TCNN is designed to capture temporal dependencies and patterns in battery data. The TCNN architecture typically consists of multiple layers of 1D temporal convolutional blocks, which are followed by pooling and fully connected layers. In order to capture long-range dependencies in the input sequences, the TCNN utilizes dilated causal convolution. By employing dilated convolutions, the model can incorporate a broader temporal context without significantly increasing computational

costs. Additionally, attention mechanisms are integrated into the TCNN to focus on relevant temporal features of battery conditions. This integration allows the model to prioritize important patterns of the battery data to enhance its ability to analyze and interpret the temporal dynamics effectively. However, according to Andrew Ng, founder of DeepLearning AI, good data are defined consistently and covers all edge cases [47].

In this paper, a novel data pre-processing model is proposed that integrates the current research findings on the SOH estimation of Li-ion batteries with the most recent advancements in DL algorithms. With this model, the numbers and usefulness of training data that are fed into the ML models are increased to improve the accuracy of the battery SOH estimation. The proposed pre-processing method converts the 1D discharge voltage data for all cycles to a 2D data set (discharge voltage cycle) extracts the new training data set from the large 2D data set by considering a constant length and width for the sliding window and moving this window over the entire 2D data set. Then, by feeding the 2D training voltage data to an ML model, features are automatically extracted in the training process. Finally, the battery voltage in the next cycle is predicted, and the SOH of the battery is calculated. An advantage of this model is that the autonomous feature extraction technique eliminates the need for extra mathematical calculations and the manual evaluation of measured data for extracting features. Another advantage is that the root-mean-squared error (RMSE) values are considerably reduced only by applying the proposed data pre-processing technique without increasing the complexity of the ML model. The performance of the proposed technique is evaluated on the NASA battery degradation data set, which is introduced in the next section.

## 3. Battery Data and Health Feature Extraction

Battery SOH is an important indicator of the health status of the battery over its lifetime that needs to be continuously monitored by the BMS. According to the definition, the SOH is the ratio of the current capacity of the battery to the capacity of the battery at the beginning of its life, i.e., the capacity of the new battery, as shown below:

$$SOH_t = \frac{C_t}{C_0} \tag{1}$$

where $C_t$ and $C_0$ represent the capacity of the battery in cycle $t$ and its initial capacity, respectively. Commonly, the SOH is represented in the form of a graph showing the trend of the battery degradation against the operation cycles (charging and discharging) of the battery. In this section, the data set used for estimating the SOH of the battery is introduced. Further, the effect of battery degradation on voltage, current, and temperature during the charging and discharging of the battery is described, and the proposed feature extraction method is explained.

### 3.1. Battery Data Set

An 18,650 rechargeable Li-ion battery with a capacity of 2.0 Ah was chosen from the NASA Ames Prognostics Center of Excellence (PCOE) open-access battery data set for the examination of the proposed method for battery SOH estimation. To collect the data set, three different operational profiles (charge, discharge, and electrochemical impedance spectroscopy) were applied to Li-ion batteries. Deep discharge aging effects were induced by setting some voltage thresholds lower than the end voltage. Also, repeated charge and discharge cycles were used to accelerate the battery aging. The tests ended when the batteries experienced a 30% capacity fade (from 2 Ah to 1.4 Ah), defining the end-of-life (EOL) criterion. The charging data were obtained using CC and CV strategies at a temperature of 24 °C. The battery was charged at 1.5 A until reaching 4.2 V, then charged at 4.2 V until the current dropped to 20 mA. Discharge data were obtained with a constant current of 2 A until the voltage dropped to 2.7 V for battery B0005, 2.5 V for battery B0006, and 2.2 V for battery B0007. The battery capacity degradation for the selected batteries is shown in Figure 2, and the battery data set is presented in Table 1.

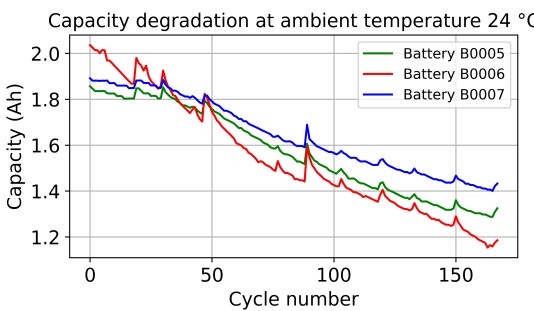

**Figure 2.** Capacity degradation for batteries B0005, B0006, B0007, and B0018 in NASA data sets.

**Table 1.** NASA data set for analyzing lithium-ion battery degradation.

| Battery ID | Rated Capacity | Rated Voltage/End Voltage | Discharge Current | CC/CV |
|------------|----------------|---------------------------|-------------------|-------|
| B0005 | 2 Ah | 3.7/2.7 V | 2A | 20 mA |
| B0006 | 2 Ah | 3.7/2.5 V | 2A | 20 mA |
| B0007 | 2 Ah | 3.7/2.2 V | 2A | 20 mA |

### 3.2. Extraction of Battery Health-Related Features

During the CC phase, a fixed current is applied to the battery until it reaches a certain voltage threshold. In the case of an aged battery, its internal resistance tends to increase over time. The higher internal resistance of an aged battery results in an increased voltage drop and reduces the effective voltage available for the CC charging of the battery. Additionally, the increased internal resistance leads to losses of more energy within the battery. These power losses manifest as heat generation, which can be detrimental to the battery's efficiency and health. Excessive heat generated during charging can accelerate the degradation of the battery, compromising its capacity and lifespan [48].

The charging and discharging profiles of voltage, current, and temperature for battery B0005 in various cycles are shown in Figure 3. As can be observed, for the aged battery, the required time to reach 4.2 V in CC is shorter, and the slope of the current in the CV charging stage is smaller as compared with the new battery (Cycle 1). Also, the temperature of the more-used batteries reaches its maximum value more quickly. According to the discharging profiles with the CC, the time of discharge and time to reach the maximum temperature are decreased, while the slope of the discharging voltage is increased.

Many studies use measured data such as the voltage, current, and temperature to predict the battery SOH. In these methods, the amount of required data for each charging and discharging cycle is too large. Therefore, efficient methods are required to reduce the size of the data or create more abstract knowledge by extracting features from the measured data without losing any important information. In addition, there are parameters such as the electrochemical reaction rate and the change in the internal resistance of the battery that are important indicators of the SOH of the battery but are not used due to practical issues and a lack of measurement devices and their poor applicability. Measuring internal resistance accurately requires specialized equipment and techniques, and this often involves the use of high-precision instruments and complex measurement setups, the availability and affordability of which may vary, especially in certain environments. Also, interpreting and applying internal resistance measurements correctly can be complex. Factors such as temperature, state of charge, and operating conditions can influence resistance readings and require careful consideration and analysis [48]. In addition, choosing features that can efficiently represent the SOH of the battery and evaluating the correlation among different variables are challenging tasks to improve the performance of the prediction methods. Because of the above-mentioned problems and the complex behavior of batteries

in different operating conditions, automatic feature extraction from measured data is highly needed.

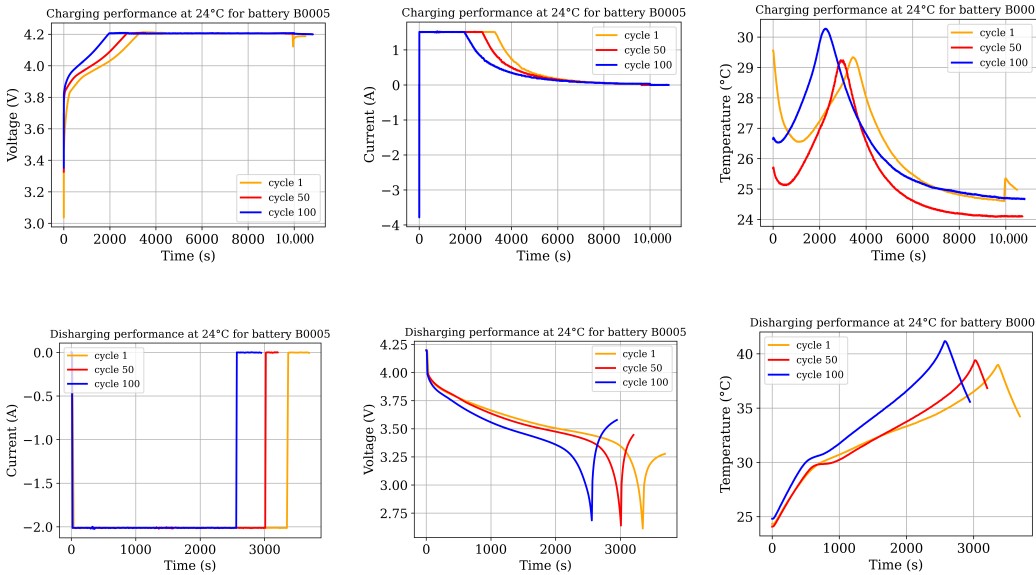

**Figure 3.** Charging and discharging voltage, current, and temperature for cycles 1, 50, and 100.

## 4. Methods

Extracting features from only one cycle of voltage data is insufficient for accurately predicting the behavior of subsequent cycles. To address this limitation, we introduced a novel approach to converting the 1D voltage data into a 2D representation using a sliding window technique. The goal is to capture the temporal patterns and dependencies within the voltage data by considering the information from previous cycles. By transforming the 1D data into a 2D matrix and applying a sliding window of a specific size (e.g., 10 previous cycles), we can extract features from a broader temporal context that allows us to consider not only the preceding cycle but also the patterns and trends observed in the previous cycles. This enables us to capture more comprehensive information and potentially improve the accuracy of predictions for future cycles. The proposed SOH estimation method consists of three stages: data pre-processing, model training, and SOH prediction using the validated model as shown in Figure 4.

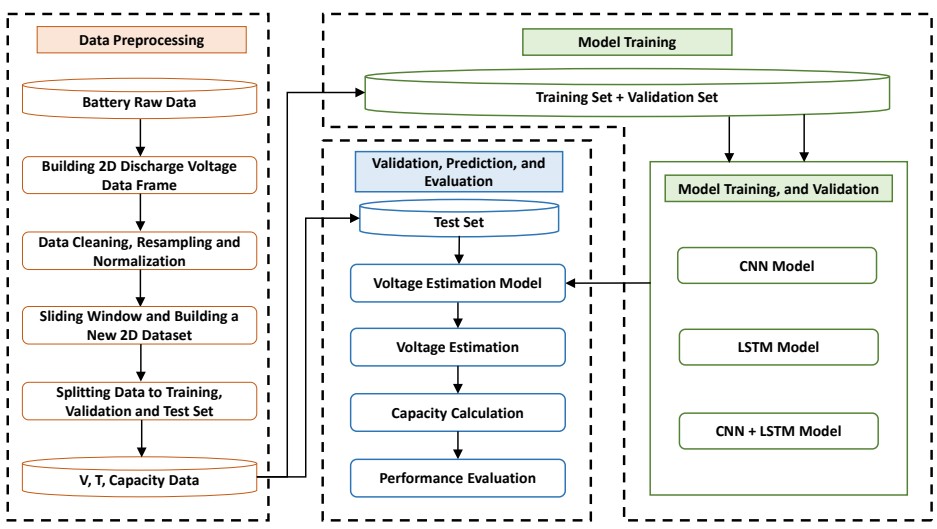

**Figure 4.** The proposed battery SOH prediction framework.

In this section, the proposed DL-based battery SOH estimation method is explained in detail. First, the approach to transform the measured time series voltage data to a 2D data set and 2D data processing is introduced. Then, the training process is presented. To verify the performance of the proposed pre-processing method, two other DL models are presented and discussed.

### 4.1. Pre-Processing Method

The measured data of the battery, including voltage, current, and temperature, are among the most widely used data for SOH estimation and are typically measured directly at constant intervals. The proposed pre-processing technique included five steps, as follows.

### 4.1.1. Cleaning Data

Data cleaning is the process of identifying and correcting inaccurate records in a data set, as well as recognizing and removing unreliable or irrelevant parts of the data. In this step of the proposed data pre-processing method, the irrelevant discharge voltage data were replaced using the interpolation technique. When the discharge voltage values declined to the end voltage, the discharge currents were interrupted, leading to the bounce-back phenomenon in the measured discharge voltages. At this step, these values of voltage data that were less than the end voltage or measured after discharge current interruption were set to the end voltage using Table 1. Figure 5a represents the cleaned data of the measured voltage at constant time intervals for the discharging of battery B0005 over 168 cycles. It is worth noting that two sampling rates of 10 and 20 s were used to collect the data in the NASA data set. Unification of data was then performed in the resampling stage.

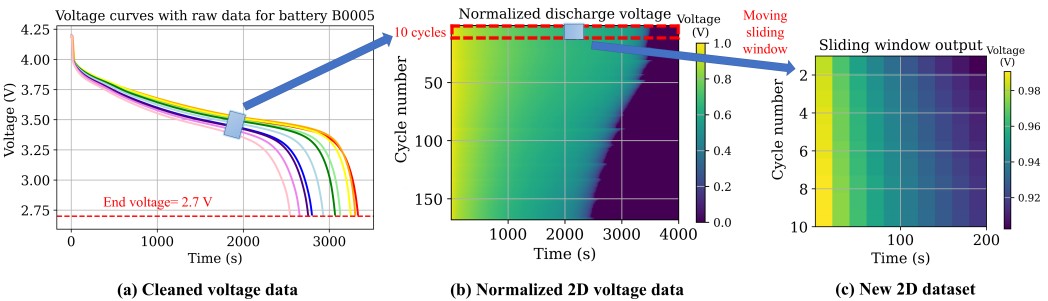

**(a) Cleaned voltage data**　　　　**(b) Normalized 2D voltage data**　　　　**(c) New 2D dataset**

**Figure 5.** Building the new training data set after cleaning (**a**), resampling, and normalization (**b**) and applying the 2D sliding window with the size of $10 \times 10$ (**c**). Processes (**a**–**c**) adhere to a 20 s voltage data sampling rate, while all three processes collectively generate a new 2D data set every 200 s with 10 points with each point representing a 20 s interval in the voltage data.

### 4.1.2. Resampling Data

By increasing the number of measured data, the number of variables in the DL model increased, which could affect the processing time and complexity of the learning process. To resolve this issue, the sampling rate was increased to have one voltage measurement every 20 s. In the initial cycles with a 10-second sampling rate, a resampling approach ws employed to average every two measured voltage data points.

### 4.1.3. Normalizing Data

Training a NN on data with different scales can lead to large weighting parameters. Such models can be poor in learning and tend to be sensitive to changes in the input values. Hence, the data were normalized in the range of [0, 1] using a min-max scaling algorithm following Equation (2):

$$X_{norm} = \frac{X - X_{min}}{X_{max} - X_{min}} \tag{2}$$

where $X_{min}$ and $X_{max}$ represent the minimum and maximum battery discharging voltage values, respectively. The minimum values, denoted as $X_{min}$, correspond to the end voltage values, as provided in Table 1. The maximum voltage values were determined by considering the highest voltage observed during the initial discharge cycles.

### 4.1.4. Building the 2D Discharge Voltage Data Set

In many battery capacity estimation methods, the collection of various features from measured data is a prerequisite. The accuracy of the estimation in these methods relies on the quality of the extracted features and how they are calculated. Also, the correlations between measured variables at different sampling periods must be analyzed to utilize all historical data efficiently. However, doing this manually with high accuracy is not a straightforward task. This issue can be resolved by using a CNNs. To employ a CNN for automated feature extraction and capacity estimation, an initial transformation process is required to generate a 2D data set from 1D-normalized voltage data. As shown in Figure 5b, 200 sequential data points with a sample rate of 20 s were extracted from the battery voltage data of each discharging cycle for all 168 cycles. Then, to transfer the 1D data to 2D data, a new matrix with the dimension of $168 \times 200$ was created from the discharging voltage data. As we can see, the battery degradation is visible in the 2D data presented as an image. The battery capacity can be calculated in each cycle using the voltage data samples.

### 4.1.5. Applying a Sliding Window and Building a New 2D Data Set

In this paper, a 2D sliding window was used to create a new data set for battery SOC and SOH estimation. The new data set was extracted from the original data set by considering a constant length and width for the window and moving this window over the entire data set. Two sampling windows with dimensions of $[10 \times 10]$ and $[1 \times 1]$ were used to build the training (X) and target (Y) data sets, respectively. With this newly formed data set, changes in voltage values within the selected time slice could be monitored using a ML model to predict the voltage in the subsequent cycle.

Figure 5c illustrates the outcomes of the proposed methodology, employing a sliding window to construct a new 2D data set. It shows a new set of training data (X) generated through resampling, normalization, and the application of a $[10 \times 10]$ sliding window to the discharge voltage data obtained from cycles 1 to 10. In this process, a new set of the training data's target value (Y) was derived by utilizing a $[1 \times 1]$ sliding window on cycle 11. To predict discharge voltage values for each cycle, an ML model ws trained using this newly constructed data set. Figure 6 explains the sequence of steps involved, including the selection of a new 2D test data from cycles 120 to 129 (a), deploying this data to the trained CNN model (b), predicting the normalized discharge voltage for cycle 130, and subsequently comparing the actual and predicted voltage values. The prediction results for cycle 130 show that the number of predicted voltage values that were more than the end voltage is 115 voltage samples with a sample rate of 20 s, and the predicted discharge time is 2300 s ($115 \times 20$).

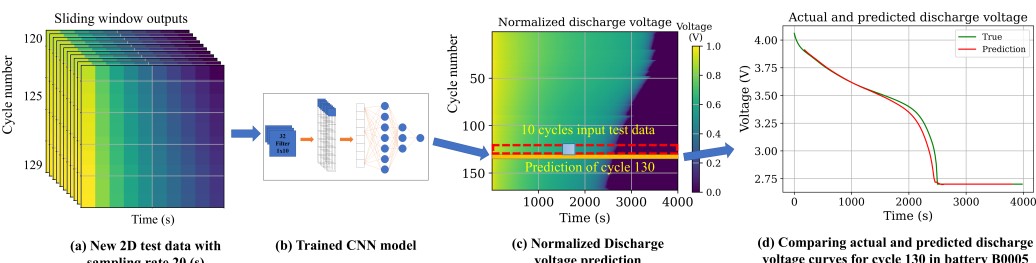

(a) New 2D test data with sampling rate 20 (s)  (b) Trained CNN model  (c) Normalized Discharge voltage prediction  (d) Comparing actual and predicted discharge voltage curves for cycle 130 in battery B0005

**Figure 6.** Discharge voltage prediction process (cycle 30): (**a**) creating new 2D test data, (**b**) deployment of the data to the trained CNN model, (**c**) prediction of normalized discharge voltage values, and (**d**) comparison between predicted and actual values.

The battery discharge time, representing the duration until the measured voltages decreased to the end voltage, was calculated using Equation (3).

$$t_{\text{cycle}_{(i)}}^{\text{discharge}}(s) = N_{\text{cycle}_{(i)}}^{\text{discharge}} \times \text{sampling rate}(s). \tag{3}$$

Table 1 provides information on the constant discharge currents for selected batteries across all cycles. It is noteworthy that, in each cycle, the batteries were discharged to the specified end voltages. The battery discharge capacity was subsequently computed using Equations (4)–(6) [49].

$$I_{\text{cycle}_{(i)}}^{\text{discharge}} = 2\ A, \tag{4}$$

$$Q_{\text{cycle}_{(i)}}^{\text{discharge}}(Ah) = I_{\text{cycle}_{(i)}}^{\text{discharge}} \times t_{\text{cycle}_{(i)}}^{\text{discharge}}(h), \tag{5}$$

$$Q_{\text{cycle}_{(i)}}^{\text{discharge}}(Ah) = I_{\text{cycle}_{(i)}}^{\text{discharge}} \times t_{\text{cycle}_{(i)}}^{\text{discharge}}(s) \times \frac{1}{3600}. \tag{6}$$

in which $Q_{\text{cycle}_{(i)}}^{\text{discharge}}$ represents the discharge capacity for cycle $i$ in ampere-hours (Ah), $t_{\text{cycle}_{(i)}}^{\text{discharge}}$ denotes the discharge time for cycle $i$ in hours (h), $I_{\text{cycle}_{(i)}}^{\text{discharge}}$ shows the discharge current for cycle $i$ in amperes (A), $N_{\text{cycle}_{(i)}}^{\text{discharge}}$ indicates the number of voltage samples that are more than the end voltage for cycle $i$, and the sampling rate shows the rate at which voltage data are sampled, measured in seconds (s).

*4.2. Training Model*

Time series data are prevalent and constantly generated in many technical processes. Thus, efficient strategies for extracting useful information from these data are critical. From AlexNet, ResNet, Inception, and Xception to the SSD and YOLO series, DNNs have made significant advances in machine vision due to their efficient feature extraction capabilities. In this paper, a DNN is used to extract features from battery data to estimate the SOH of the battery. As can be seen in Figure 2, the general trend indicates a definite decline in the Li-ion battery capacity. As a result, a model that takes gradual degradation into account has certain inherent advantages.

4.2.1. CNN Model for Automatic Feature Extraction

CNNss are a class of artificial NNs used in DL for automatic feature extraction from high-dimensional data, which generally is implemented on 2D data (e.g., ImageNet, object detection, segmentation, medical imaging, diagnosis, etc.) [50]. CNNss have been employed in various applications, such as image processing, classification, and natural language processing. Further, CNNs have been used for time series forecasting and estimation, as well as fault diagnosis [51]. For each influencing input element of the model, a CNN can perform feature extraction and data dimension reduction, which helps to reduce the number of feature parameters and model complexity [52].

CNNss also have structural properties, such as local correlation and weight sharing, which serve to prevent overfitting in the model [53]. Time series data are prevalent and constantly generated in many technical processes. Thus, efficient strategies for extracting useful information from these data are critical. Considering the merits of CNNs, such as automatic feature extraction and a low risk of overfitting, their applications in dealing with a large number of time series signals have also been investigated. This is one of the main reasons for selecting the CNN model for automatic feature extraction and capacity prediction for battery degradation in this paper. To estimate the battery SOH, a convolution layer was used to extract features from the input data. In the proposed method, the size of the input training data and convolution kernel (a kernel is a matrix of weights that is multiplied by the input to extract features) were set to [10 × 10] and 10, respectively. After

scanning all the input data by moving the convolution kernel, the output features were obtained. The 1D convolution is presented in Figure 7, as well as Equations (7) and (8):

$$s_{(n)} = f_{(n)}g_{(n)}, \tag{7}$$

$$s_{(n)} = f_{(n)}\sum_{i=1}^{l-1} g_{(n-m)}. \tag{8}$$

where $s_{(n)}$ is the convolution result, $f_{(n)}$ is the input data, and $g_{(n)}$ is the convolution kernel with the size of $l$.

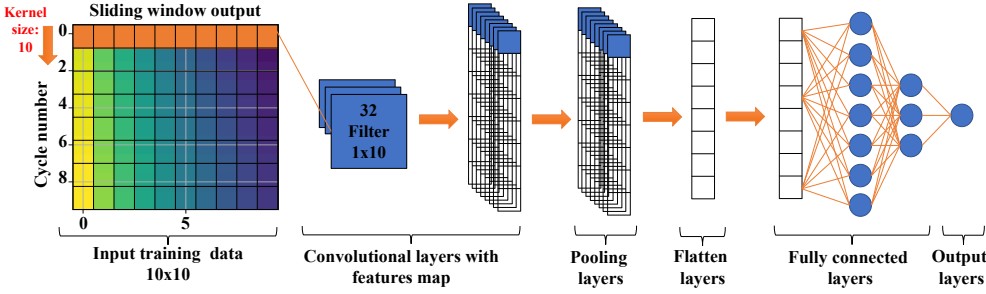

**Figure 7.** 1D convolution NN for automatic feature extraction from battery voltage data.

### 4.2.2. LSTM Model for Temporal Dependency

RNNs are a popular type of DL architecture, where connections between units form a directed graph along with the sequence of information from the input. RNNs process a sequence of input data by using their internal states. RNNs suffer from the vanishing gradient problem, which has a major negative effect on their accuracy. An enhanced version of the RNN architecture is LSTM, which overcomes the vanishing gradient problem via the concept of gates (input, forget, and output gates) and memory cells. LSTM operation is defined by Equations (9)–(14) [30].

$$f_t = \sigma\left(\widehat{W}_f \cdot [h_{t-1}, x_t] + B_f\right), \tag{9}$$

$$i_t = \sigma\left(\widehat{W}_i \cdot [h_{t-1}, x_t] + B_i\right), \tag{10}$$

$$\hat{C}_t = \tanh\left(\widehat{W}_C \cdot [h_{t-1}, x_t] + B_C\right), \tag{11}$$

$$C_t = (f_t \times C_{t-1}) + (i_t \times \hat{C}_t), \tag{12}$$

$$o_t = \sigma\left(\widehat{W}_o \cdot [h_{t-1}, x_t] + B_o\right), \tag{13}$$

$$h_t = o_t \times \tanh(C_t). \tag{14}$$

in which $x_t$ is the network input, $h_t$ is the output of the hidden layer, $o_t$ represents the output gate, $\sigma$ represents the Sigmoid function, and $C_t$ is the cell state (memory). The state candidate's values are represented by $\hat{C}_t$, and $\widehat{W}_f$, $\widehat{W}_t$, $\widehat{W}_c$, and $\widehat{W}_o$ are the weights for the input, output, forget gate, and memory cells, respectively. $B_f$, $B_i$, $B_c$, and $B_o$ represent the bias for the input, output, forget gate, and cell, respectively. The input gate decides whether input data will be reserved or not, while the forget gate verifies if data will be lost or not, and the cell records the processing state. Finally, the output is delivered through the output gate. This architecture is specially designed to address the vanishing gradient problem in RNNss.

The LSTM structure is shown in Figure 8.

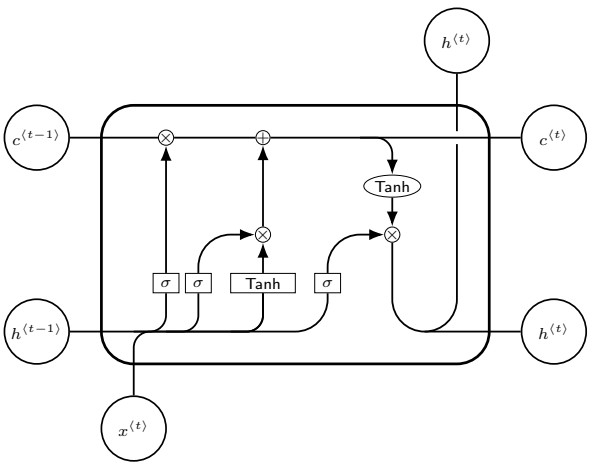

**Figure 8.** LSTM model.

### 4.2.3. CNN–LSTM Model for Automatic Feature Extraction

The necessity of dealing with the vast volumes of sequential sensor data that are connected to the battery is one of the obstacles that arise when trying to predict the SOH of a battery. It has been shown that CNN–LSTM models have the potential to be effective in addressing these issues, and they are well-matched to succeed in accomplishing this objective [32]. In Figure 9, a CNN–LSTM model for automatic feature extraction is illustrated. The model comprises two primary components: a CNN and LSTM. The CNN consists of a convolutional layer followed by a max-pooling layer. The convolutional layer has 32 filters with a size of [10 × 1]. The max-pooling layers reduce the height dimension. The LSTM, with 128 hidden units, is responsible for capturing the temporal dependencies in the data. The output of the LSTM layers is then fed into a fully connected layer with a single neuron, which produces the output value. The CNN–LSTM model is trained using a mean squared error loss function. Compared to LSTM, the CNN can extract spatial information from the input data, such as measured voltage and temperature. On the other hand, LSTM can capture the temporal dependencies in the training data.

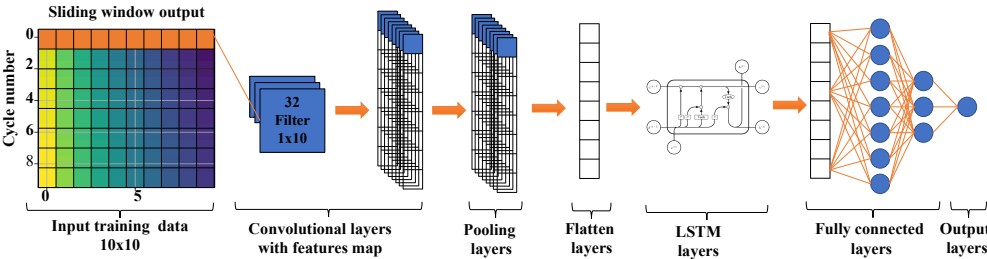

**Figure 9.** CNN–LSTM model.

### 4.3. Performance Evaluation

The proposed method was evaluated by using two standard metrics: mean absolute error (MAE) and RMSE. The mathematical formulas of these metrics are given in the Equations (15) and (16), where $y_i$ and $\hat{y}_i$ are the actual and predicted values, respectively.

$$MAE = \frac{1}{n}\sum_{i=1}^{n} |y_i - \hat{y}_i|, \tag{15}$$

$$RMSE = \sqrt{\frac{1}{n}\sum_{i=1}^{n} (y_i - \hat{y}_i)^2}. \tag{16}$$

## 5. Results and Discussion

In this section, the ML-based model for predicting the SOH of the battery with the proposed pre-processing method is applied to the data set provided by NASA for the degradation analysis of Li-ion batteries. In the following, we present the simulation configuration and establish various scenarios for conducting comparative analyses.

### 5.1. Simulation Setup

The battery SOH estimation model was trained on a GPU and implemented in Python (V3.9) with TensorFlow and Adam as the optimizer with mini-batch gradient descent. For quick demonstration purposes, the model was trained for 10 epochs using the 2D sliding windows with a size of $10 \times 10$.

### 5.2. Comparative Analysis

In this paper, B0005, B0006, and B0007 batteries were selected for the simulation analysis (following the numbering used in the NASA battery data set). From the available data for each battery, 60, 80, and 100 data points were used for training; 20 data points were used for validation, and the remaining data were used for testing. To evaluate the effectiveness of the proposed SOH estimation framework, the following four cases were investigated:

1. Case 1: with and without data pre-processing;
2. Case 2: with data pre-processing and different sizes of training data;
3. Case 3: with data pre-processing and different sizes of the sliding window;
4. Case 4: with data pre-processing and with dropout.

5.2.1. Case 1: Battery SOH Estimation with and without the Proposed Data Pre-Processing Method

The performance of the proposed method for SOH estimation was verified using three Li-ion batteries: the B0005, B0006, and B0007 batteries. The first 100 cycles of each battery were used as training data, while the following cycles were used for the test data. Also, 20 cycles of the training data were chosen for validation. In Figure 10, the difference between the actual and estimated capacities and the error values for the test data are shown. Figure 10 shows how well the CNN–LSTM model with the proposed pre-processing method estimates the capacity degradation trends of the selected batteries. According to the obtained results, the overall estimation error is below 5% for all three batteries. The capacity regeneration phenomena generate the peak values of the irregularities in real capacity degradation profiles, where errors of more than 1% are observed. Rest time in batteries can trigger certain electrochemical processes that partially reverse the degradation effects, resulting in an increase in battery capacity and influencing the overall degradation trend. These processes, occurring during periods of rest, can mitigate the detrimental effects on the battery's performance and extend its lifespan [54].

In Table 2, RMSE and MAE values for SOH estimation are provided with CNN and LSTM models with and without the proposed data pre-processing method, considering the capacity data in each cycle as training data. The results show that the proposed pre-processing model improves the accuracy of CNN and LSTM models for SOH estimation.

In Figure 11, the training and validation loss per epoch for the CNN-LSTM model applied to the battery B0007 data set are presented. As evidenced in the figure, both the training and validation loss values drop below 0.001 after just 4 epochs. Consequently, guided by this observation, the decision was made to set the number of epochs for simulation in all cases to 10. It is noteworthy that, for all simulations, hyperparameter tuning was employed on the ML model. This practice was implemented to optimize the performance of SOH estimation for batteries.

**Table 2.** Battery SOH estimation errors in different scenarios.

| Battery ID | Error Type | With Proposed Pre-Processing | | Without Proposed Pre-Processing | |
|---|---|---|---|---|---|
| | | **CNN** | **LSTM** | **CNN** | **LSTM** |
| B0005 | RMSE | **0.010** | 0.012 | 0.061 | 0.084 |
| | MAE | **0.007** | 0.008 | 0.054 | 0.071 |
| B0006 | RMSE | **0.013** | 0.014 | 0.052 | 0.093 |
| | MAE | **0.009** | 0.009 | 0.041 | 0.079 |
| B0007 | RMSE | **0.009** | 0.010 | 0.076 | 0.088 |
| | MAE | **0.006** | 0.006 | 0.072 | 0.081 |

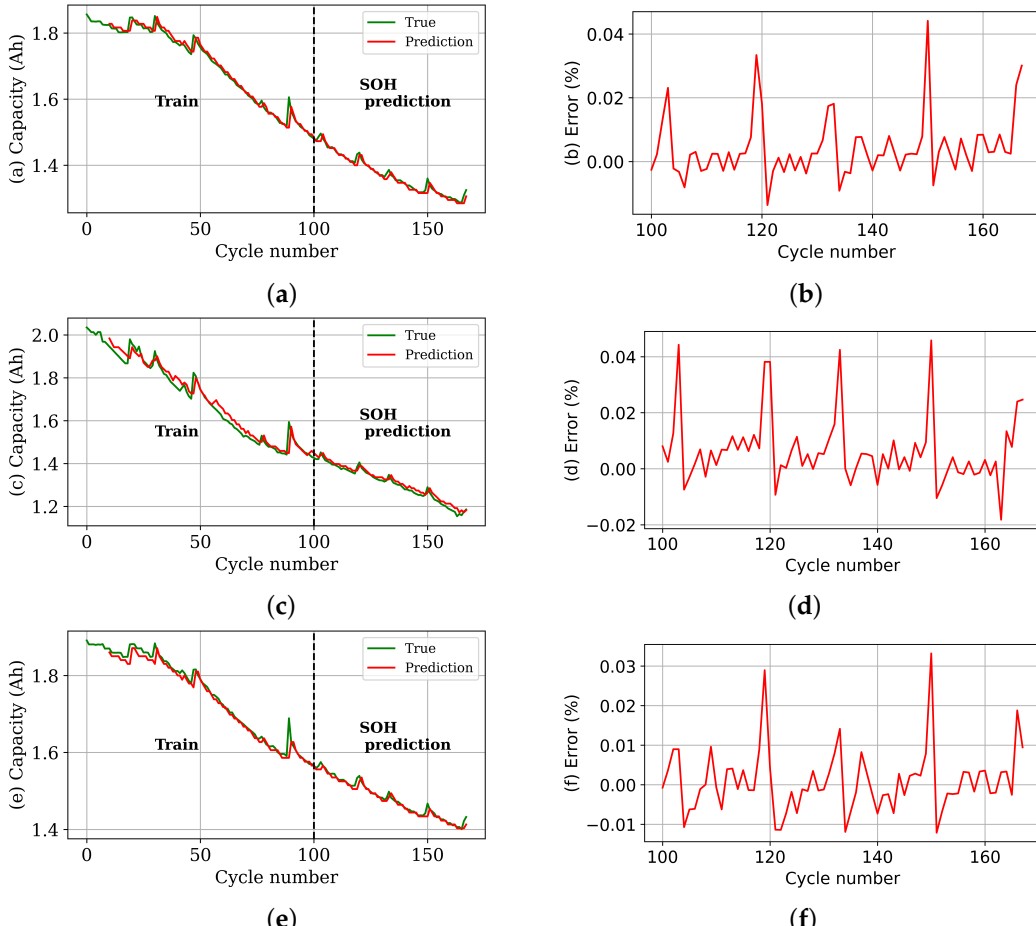

**Figure 10.** Evaluation of the proposed pre-processing method's performance in SOH prediction for the subsequent cycles of battery B0005 (**a**,**b**), B0006 (**c**,**d**), and B0007 (**e**,**f**) using the CNN–LSTM model.

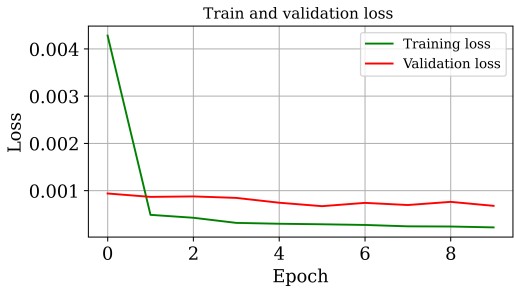

**Figure 11.** Training and validation loss per epoch for the CNN–LSTM model on the battery B0007 data set.

5.2.2. Case 2: Battery SOH Estimation with the Proposed Data Pre-Processing Method and Using Different Sizes of Training Data

In this section, we compare the RMSE values for different sizes of training data to investigate the effect of the training data size on the performance of the proposed method. In this analysis, CNN, LSTM, and CNN–LSTM models, which are often utilized in computer vision and time series data processing, were investigated. Different training data sets with 80, 100, and 120 cycles were used to train the models, and the remaining data were used for testing. Similar to the previous case, 20 cycles of each training set were continually chosen for validation. The RMSE values of the three selected models on the B0005, B0006, and B0007 data sets are shown in Table 3. As can be seen, by decreasing the number of training data from 120 to 80, the performance of all models is gradually degraded, while all error values are less than 0.014. The CNN-LSTM model has the best accuracy.

**Table 3.** RMSE of the prediction results using different sizes of training data.

| Battery ID | Training Data | Model | | |
|---|---|---|---|---|
| | (cycle) | CNN | LSTM | CNN-LSTM |
| | 1–80 | 0.014 | 0.016 | 0.014 |
| B0005 | 1–100 | 0.011 | 0.012 | 0.011 |
| | 1–120 | 0.010 | 0.011 | **0.010** |
| | 1–80 | 0.021 | 0.023 | 0.021 |
| B0006 | 1–100 | 0.014 | 0.013 | **0.012** |
| | 1–120 | 0.013 | 0.013 | 0.014 |
| | 1–80 | 0.016 | 0.015 | 0.014 |
| B0007 | 1–100 | 0.009 | 0.010 | 0.008 |
| | 1–120 | 0.008 | 0.010 | **0.007** |

5.2.3. Case 3: Battery SOH Estimation with the Proposed Data Pre-Processing Method and Using Different Sliding Window Sizes

Table 4 shows the RMSE for the three selected battery data sets and fixed window sizes of 5, 10, and 20. It can be observed that small window sizes of 5 and 10 yield the minimum RMSE values. For the CNN model, the minimum RMSE is obtained for a window size of 5, whereas the LSTM and CNN–LSTM models produce the minimum error with a window size of 10. In general, smaller window sizes show better results. However, by decreasing the size of the sliding window, the number of variables is increased. Therefore, the window size of 10 is the best size for the proposed pre-processing method for battery SOH prediction, which was obtained by trial and error. As a result of increasing the dimension of the window size to more than 20, the dimension of the input data for ML and the error of prediction would also be increased. Therefore, values of more than 20 were not considered for analysis in this paper.

**Table 4.** RMSE of the prediction results using different sliding windows.

| Battery ID | Sliding Window | Model | | |
|---|---|---|---|---|
| | | CNN | LSTM | CNN-LSTM |
| | 5 × 5 | **0.009** | 0.012 | 0.010 |
| B0005 | 10 × 10 | 0.010 | 0.012 | 0.010 |
| | 20 × 20 | 0.011 | 0.017 | 0.012 |
| | 5 × 5 | **0.012** | 0.014 | **0.012** |
| B0006 | 10 × 10 | 0.014 | 0.013 | 0.012 |
| | 20 × 20 | 0.014 | 0.014 | 0.014 |
| | 5 × 5 | 0.010 | 0.009 | **0.008** |
| B0007 | 10 × 10 | 0.009 | 0.010 | 0.009 |
| | 20 × 20 | 0.011 | 0.009 | 0.010 |

### 5.2.4. Case 4: Battery SOH Estimation with the Proposed Data Pre-Processing Method and Using Data Dropout

In this case, the focus is on battery SOH prediction using different models with dropout. To improve the accuracy and reduce the risk of overfitting, dropout was applied as a regularization technique to the estimation models. Dropout was performed by randomly choosing nodes to be dropped out with a given probability at each weight-updating iteration. The performance of the models was evaluated by calculating the RMSE values. Table 5 shows that dropout improves performance in the LSTM and CNN-LSTM models but increases error values in the CNN model when reducing training size. The best results are highlighted in the table. Overall, the proposed data pre-processing technique and dropout incorporation contributed to accurate SOH estimation, benefiting battery health monitoring and predictive maintenance practitioners. As mentioned before, the focus of this paper is on predicting the battery voltage in the next cycle and calculating the SOH of the battery. However, for an effective RUL estimation, longer time-frame estimations are needed. Extending the proposed approach for longer-term predictions of the SOH of Li-ion batteries by incorporating additional factors to enhance the accuracy of RUL estimation will be considered by the authors in their future works. This may involve utilizing more advanced ML algorithms and considering a broader range of features to capture the complex degradation patterns exhibited by Li-ion batteries over time.

**Table 5.** RMSE of the prediction results with and without dropout.

| Battery ID | Dropout | Model | | |
|:---:|:---:|:---:|:---:|:---:|
| | | **CNN** | **LSTM** | **CNN-LSTM** |
| B0005 | NO | 0.011 | 0.012 | 0.011 |
| | YES | 0.019 | 0.012 | **0.10** |
| B0006 | NO | 0.014 | 0.013 | 0.014 |
| | YES | 0.013 | **0.012** | 0.013 |
| B0007 | NO | 0.009 | 0.010 | 0.009 |
| | YES | 0.010 | 0.009 | **0.008** |

## 6. Battery DT with Online Learning

The conventional methods of battery SOH estimation involve building a model from sensor data and other sources and applying that model to estimate the behavior of the battery. However, this method is not completely capable of capturing the complicated and dynamic behavior of batteries in real-world applications and their aging mechanisms that are necessary for developing the battery DT. In this regard, adopting online learning methods is crucial. Online learning is a method that uses ML algorithms to continuously update and improve the accuracy of the prediction model. By embedding online learning into the SOH estimation model, the model is continuously updated with new data as it becomes available, enabling it to improve over time. In this way, the real-time data stream is continuously analyzed and the model parameters are updated in real-time employing ML techniques, which provides a more accurate representation of the behavior of the battery. The estimation of battery SOH with online learning is shown in Figure 12. These models can recognize and respond to changes in the behavior of batteries over time, which is especially crucial for battery usage in mobile applications, for instance in electric vehicles, where the battery undergoes a wide range of operating conditions and usage patterns.

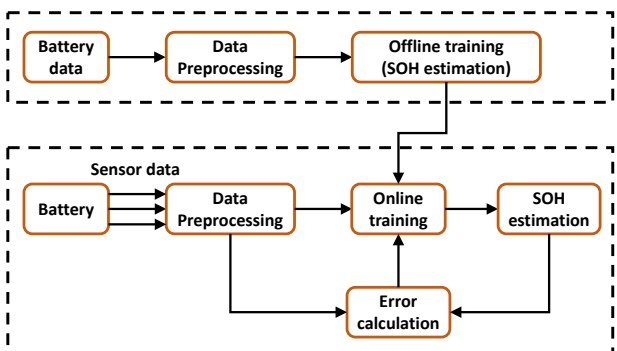

**Figure 12.** Battery SOH estimation with online learning.

The ability to continually learn, adjust over time, and avoid catastrophic forgetting are among the important challenges of online learning. The learning model must balance *stability* (the ability to keep previous information) with *plasticity* (the flexibility to adjust to new information) over time. When learning a new activity, excessive plasticity might lead to the forgetting of previously learned information. Moreover, learning sequential tasks may become more challenging due to excessive stability [55]. Li et al. [56] presented a combined CNN and transfer learning model to extract features automatically and fine-tune a pre-trained model for online battery SOH estimation. Yan et al. [57] suggested an XGBoost-based online platform for online battery health diagnosis. Data aggregation and feature extraction for real-time battery data during charging are performed using dynamic time warping for clustering measured data, and a SOH prediction model is created by using the XGBoost method.

Incorporating online learning mechanisms into the battery SOH estimation model facilitates developing a battery DT that mirrors the true condition of the battery over its lifetime, which is the scope of the future research of the authors. The proposed data pre-processing method in this paper, which can automatically extract features from raw data, is the first step toward developing battery DT.

## 7. Conclusions

In this paper, a novel data pre-processing method for the SOH estimation of Li-ion batteries was proposed. The only data that re required for SOH estimation are the discharging voltage data of the battery. In the proposed pre-processing method, the 1D discharge voltage data are used to create the 2D training data set. By using the 2D voltage data as training data in the DL model for battery SOH estimation, voltage features are automatically extracted in the training process. Three battery data sets were utilized to evaluate the performance of the proposed model and examine the impact of various factors, such as the size of the training data and the sliding window, on the SOH estimation accuracy. Further, the impact of applying the data dropout technique to the estimation method was investigated. According to the results, the proposed CNN–LSTM model provides the highest accuracy, where the maximum RMSE values of this model are lower than 1.5% for all scenarios. Moreover, the RMSE values could drop to 1% by selecting an appropriate sliding window, including dropout in the ML models, and increasing the amount of training data.

Compared to conventional approaches, which rely on battery capacity or manually extracted features from voltage and current curves, the proposed model automatically extracts features from 10 voltage curves simultaneously. It effectively monitors changes in voltage curves to predict battery SOH with a notably small RMSE value. In the future, developing battery DT by employing a hybrid model to integrate the proposed data-driven models with a model-based method for battery SOH estimation and extending the proposed method to incorporate additional factors by capturing complex degradation patterns and including longer-term SOH predictions to enhance the accuracy of RUL estimation will be considered by the authors. To do so, advanced online learning techniques will be deployed

to continuously update the prediction model using a real-time data stream and preserve the accuracy of the prediction method over the lifetime of the battery.

**Author Contributions:** Conceptualization, V.S., N.B., J.C.V., and J.M.G.; methodology, V.S. and N.B.; software, V.S. and N.B.; validation, V.S. and N.B.; formal analysis, V.S. and N.B.; investigation, V.S. and N.B.; resources, V.S., N.B., J.C.V., and J.M.G.; data curation, V.S., N.B., J.C.V., and J.M.G.; writing—original draft preparation, V.S. and N.B.; writing—review and editing, V.S., N.B, J.C.V., and J.M.G.; visualization, V.S. and N.B.; supervision, J.C.V. and J.M.G.; project administration, J.C.V. and J.M.G.; funding acquisition, J.C.V. and J.M.G. All authors have read and agreed to the published version of the manuscript.

**Funding:** This work was supported by VILLUM FONDEN under the VILLUM Investigator Grant (no. 25920): Center for Research on Microgrids (CROM).

**Data Availability Statement:** Study data available upon request from the corresponding author.

**Conflicts of Interest:** The authors declare no conflict of interest.

## Abbreviations

The following abbreviations are used in this paper:

| | |
|---|---|
| NASA | National Aeronautics and Space Administration |
| Li-ion | Lithium-ion |
| BMS | Battery management system |
| SOH | State-of-health |
| SOC | State-of-charge |
| RUL | Remaining useful life |
| EOL | End-of-life |
| DT | Digital twin |
| RBF | Radial basis function |
| MC | Monte Carlo |
| DTV | Differential thermal voltammetry |
| CC | Constant current |
| CV | Constant voltage |
| 1D | One-dimensional |
| 2D | Two-dimensional |
| ML | Machine learning |
| DL | Deep learning |
| NN | Neural network |
| FNN | Feed-forward neural network |
| DNN | Deep neural network |
| CNN | Convolutional neural network |
| TCNN | Temporal convolutional neural network |
| RNN | Recurrent neural network |
| LSTM | Long short-term memory |
| GRU | Gated recurrent unit |
| DBN | Deep belief network |
| RMSE | Root-mean-squared error |
| MAE | Mean absolute error |
| MAPE | Mean absolute percentage error |
| PSO | Particle swarm optimization |

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
