# Peer review of "Battery State-of-Health Estimation: A Step towards Battery Digital Twins"

_electronics, doi:10.3390/electronics13030587_

Round 1

Reviewer 1 Report

Comments and Suggestions for Authors

Review Comments

The introduction is comprehensive and the proposed method is interesting and novel. However the writing needs to be more concise and the presentation of the data presentation and training procedure need to be improved in order to further support the result. Below are some detailed suggestions: 

  1. Section 4.1 needs to be more concise and the figures need to be more clear. A few suggestions: First of all, merge fig5 and fig 6 into one figure. You only need Fig5a, Fig 6b&c. cleaning, resampling, normalization are common practice in such kind of data processing, so they can mentioned briefly. Second of all, It is not straightforward to explain the idea of “transfering 1D-data to 2D-data”. Suggest plot only 10 cycles in Fig5a and draw an arrow spanning from Fig5a to Fig 5b(6b) to indicate how one of the 1D voltage is represented in 2D data. In addition, please make the X-axis consistent in Fig 5a and  Fig 5b(6b) (terms like “standardized grid” are easier to understand).

  2. Section 4.1. Data needs more information. 1) Please explicitly point out the value of M and N. 2) It is confusing how you ended up with a 10X10 training X (line 399) by using a sliding window of 10X10 on original data (line 376). 3) how many 2-D data are used for the training?

  3. Section 4.1. Merge line 388-396 into previous section where the author introduce CNN. No need to introduce again.

  4. Section 4.1. Explain why sampling windows [10X10] and [1X1] are used.

  5. Consider put the network structure (fig 7 along with Equation (3) & (4)) into Supplementary information given the length of the article.

  6. It is not clear how CNN and LSTM are combined as given in Figure 9. Please be more specific in the network structure.

  7. Training procedure including the parameter tuning needs to be described in the supplementary information. Please justify the use of 10 epochs. 

  1.  
  2.  

Author Response

Manuscript No.: electronics-2710620

Title: State-of-Health Estimation: A Step Towards Battery Digital Twins

Dear reviewer of the Journal of MDPI

We would like to express our sincere gratitude to the reviewer for their careful reviews of our manuscript, and the constructive recommendations. Considering the received comments and suggestions, we have revised the original manuscript carefully. In what follows, we indicate how your comments and suggestions have been considered in the revised manuscript.

For clarity, we will quote these comments/suggestions in Black color and the authors’ answers are marked in Blue color. Following the instruction, both the original version with changes highlighted, which will be referred to as the marked-up manuscript from now on, and the clean revised version are submitted. In the marked-up manuscript, to track the changes in the original document, the changes are highlighted with yellow color.

With Best Regards,‎

The Authors.

Reviewer 2 Report

Comments and Suggestions for Authors

The review is in the attached file.

Author Response

(The authors gave the same response as above.)

Reviewer 3 Report

Comments and Suggestions for Authors

This article delves into the issue of a novel data pre-processing method for SOH estimation of Li on batteries, providing new ideas and references for academic research and practical applications in this field. Some following problems remain.

(1) Inadequate explanation of converts the 1D-discharge voltage data for all cycles to 2D-data set.

(2) The problem of CNN-LSTM model for automatic feature extraction discussed in this paper is unclear, write one section to define this.

(3) The proofs cited in this article to demonstrate various arguments about NASA lithium-ion batteries degradation are relatively few and insufficient to support the entire conclusion. Authors should provide more proofs.

(4) Strengthen citation of the latest research results to better reflect the LIB developments, such as Small 2023, 19, 2206563. Chem. Commun., 2023,59, 13313.

Comments on the Quality of English Language

The specific modification requirements include: more accurate and clear expression of some viewpoints in the text; Check and organize the references, etc.

Author Response

(The authors gave the same response as above.)

Round 2

Reviewer 1 Report

Comments and Suggestions for Authors

Review V2 2024/01/03

The author made necessary adjustment and improved the overall quality of the manuscript. However, there is still places needs to be improved.

Figure 5 is critical in understanding your 2D data approach. A few more comments to it.

  1. Clearly label each subplot with (a), (b), or (c).

  2. The x-axis in (b) is actually temporal information. Consider use “time” instead of “voltage samples”. Explain accordingly in the main text.

  3. describe how voltage is normalized, what is your X-max, X-min.

  4. In (a) voltage bounces back above 3V after hitting 2.7V threshold, but this bounce back wasn’t reflected in (b). Did you choose different time window for extracting the voltage? Provide more details from line 357 to 360.

  5. In (c). How is data resampled within the windowed? Aeraged? max pooled? please explain in line 370 and on.

  6. According to Fig 5 and line 369-374, the study intends to use discharging voltage data for first 10 cycles (training X) to predict the 11th cycle (target Y). Please validate why is this prediction important. This is critical since the real value of such approach emerge when predicting SOC beyond 100 cycles.
